# Adjuvant Immunotherapy After Resected Melanoma: Survival Outcomes, Prognostic Factors and Patterns of Relapse

**DOI:** 10.3390/cancers17010143

**Published:** 2025-01-05

**Authors:** Sergio Martinez-Recio, Maria Alejandra Molina-Pérez, Eva Muñoz-Couselo, Alberto R. Sevillano-Tripero, Francisco Aya, Ana Arance, Mayra Orrillo, Juan Martin-Liberal, Luis Fernandez-Morales, Rocio Lesta, María Quindós-Varela, Maria Nieva, Joana Vidal, Daniel Martinez-Perez, Andrés Barba, Margarita Majem

**Affiliations:** 1Department of Medical Oncology, Hospital de la Santa Creu I Sant Pau, 08025 Barcelona, Spain; 2Department of Medicine, Universitat Autònoma de Barcelona, 08193 Barcelona, Spain; 3Department of Medical Oncology, Hospital Universitario Vall d’Hebron, Vall d’Hebron Instute of Oncology (VHIO), 08035 Barcelona, Spain; 4Department of Medical Oncology, Hospital Universitario de Getafe, 28905 Madrid, Spain; 5Department of Medical Oncology, Hospital Clinic, 08036 Barcelona, Spain; 6Department of Medical Oncology, Institut Català d’Oncologia ICO, L’Hospitalet de Llobregat, 08908 Barcelona, Spain; 7Department of Medical Oncology, Hospital Universitari Parc Taulí, 08208 Sabadell, Spain; 8Department of Medical Oncology, Hospital Universitario de A Coruña (CHUAC), 15006 Coruña, Spain; 9Department of Medical Oncology, Hospital del Mar-Parc Salut Mar, 08003 Barcelona, Spain; 10Department of Medical Oncology, Hospital Central de Asturias (HUCA), 33011 Oviedo, Spain

**Keywords:** melanoma, adjuvant immunotherapy, survival outcomes, prognostic factors, patterns of relapse

## Abstract

Our study evaluates how well anti-PD-1 immunotherapy works for melanoma patients in routine clinical practice, as previous clinical trials may not fully reflect daily outcomes. We focused on patients who had melanoma surgically removed and then received anti-PD-1 therapy to prevent the cancer from returning. By reviewing data from 245 patients from several centers, we confirmed that real-world survival rates are lower than those reported in trials and identified factors that may influence recurrence, such as the location of the original primary tumor or delays in initiating therapy after surgery. This highlights the importance of real-world studies and provides insight into current needs to improve patient outcomes.

## 1. Introduction

Cutaneous melanoma accounts for 325,000 new diagnoses worldwide, with age-standardized incidence rates ranging from 14 (dark-skinned populations) to 42 (fair-skinned populations) per 100,000 person-years [1]. The prognosis of patients diagnosed with advanced melanoma has dramatically improved since the introduction of anti-PD-1-based immunotherapy and targeted therapy with BRAF and MEK inhibitors for BRAF-mutated patients [2,3,4,5].

These treatments have also improved outcomes after complete resection in the adjuvant setting. Previously, ipilimumab improved recurrence-free survival (RFS), distant metastasis-free survival (DMFS) and overall survival (OS) compared with placebo in stage III, although 7-year RFS rate was 39.2% [6]. Subsequently, nivolumab showed better RFS and DMFS with less toxicity compared with ipilimumab in stage III, while pembrolizumab improved RFS and DMFS compared with placebo [7,8,9,10,11,12]. More recently, both pembrolizumab and nivolumab improved RFS compared with placebo in stage II, although longer follow-up is needed to determine other outcomes [13,14]. Last, the combination of ipilimumab and nivolumab has also shown superior efficacy in stage IV but not in stage III [15,16]. Regarding targeted therapy, dabrafenib and trametinib also improved RFS and DMFS compared to placebo in BRAF-mutated stage III melanoma [17].

Despite these advances, recurrence rates in clinical trials are far from ideal, with 50% of patients experimenting recurrences 5–7 years [9,12]. Moreover, a review of real-world studies suggests that this rate may be higher, indicating discrepancies between real-world practice and the outcomes observed in clinical trials [18].

Several real-world studies have reported on survival outcomes and treatment of early relapse in patients with resected stage III melanoma treated with adjuvant anti-PD-1 monotherapy [18,19,20]. However, there is limited evidence regarding prognostic factors and patterns of relapse. Additionally, little is known about patients in real-world settings who have stage IIB-IIC or IV melanoma treated with anti-PD-1-based combination immunotherapy.

This study aims to describe post-treatment and survival outcomes for melanoma patients who have undergone complete resection followed by adjuvant immunotherapy in a real-world setting. Additionally, it also seeks to evaluate prognostic factors influencing these outcomes and to describe patterns of relapse observed in this patient population.

## 2. Materials and Methods

### 2.1. Study Design

This is an observational retrospective multicenter study that collected data from 8 different Spanish sites. The study protocol was approved by the institutional ethics review board, and all patient data were anonymized upon entry into the database.

Eligible patients were adults (18 years or older), with a histologically confirmed melanoma with resection of all known disease lesions, and had received at least one dose of adjuvant anti-PD-1-based therapy (either as monotherapy or in combination with another immunotherapy). Patients with less than nine months of follow-up post-treatment initiation or with uveal melanoma were excluded from the study.

Patients were enrolled consecutively from local registries if they met all the inclusion criteria and no exclusion criteria. The sample size was not restricted due to the exploratory nature of the study.

### 2.2. Outcomes and Asessments

Collected data included patient demographics and disease characteristics at the start of adjuvant therapy, details of the treatment regimen and its timing, observed toxicities, pattern of relapse and survival outcomes. Adverse events were graded using the Common Terminology of Cancer Adverse Events (CTCAE) criteria, version 5.0.

Locoregional relapses were defined as those occurring at the previous primary surgical site, regional skin, subcutaneous tissue or lymph nodes, while distant relapses were defined as those occurring at any other location, even if detected at a previously resected metastatic site.

Recurrences were classified as early relapse (primary resistance) if they occurred during adjuvant treatment or within 12 weeks after treatment completion, in accordance with the Society for Immunotherapy in Cancer (SITC) criteria [21].

Survival times were calculated from the start date of therapy and the occurrence of the following events: recurrence-free survival (RFS), defined by the date of first recurrence or death; distant metastasis-free survival (DMFS), marked by the first distant recurrence or death; time to next treatment (TTNT), defined by the date of initiation of the next systemic therapy for melanoma or death; and overall survival (OS), defined by the date of death. If a patient’s date of death was not documented, survival times were censored at the last recorded date the patient was known to be alive.

### 2.3. Statistical Analysis

Descriptive statistics were used to assess baseline demographics characteristics, disease-related variables and treatment-related data. For continuous variables, results are reported as median with ranges, while categorical measures are summarized by patient counts and percentages.

The Kaplan–Meier method was employed to estimate median survival times (with 95% confidence intervals, CI) for RFS, DMFS, TTNT and OS, as well as to generate survival curves. Survival rates at 18 and 36 months (with 95% CI) were calculated using an actuarial method. The reverse Kaplan–Meier method was used to determine the median follow-up time and descriptive statistics were used to report the minimum follow-up duration and range.

The log-rank test was used to compare survival times based on different clinical subgroups, and the single-variable Cox proportional hazards regression method was used to calculate univariate hazard ratio (HR). For time variables where median survival was not reached, subgroup comparisons using the log-rank test and HR were considered informative if each subgroup had 10% or more of events. For clinical subgroups with a significant association with survival, a multivariate Cox regression analysis was conducted. A time-dependent Cox regression analysis was conducted to evaluate the relationship between toxicity and survival, adjusting potential biases due to different treatment exposure.

A significance threshold of *p* < 0.05 was set for all analyses, which were conducted using SPSS (PAWS statistical software), version 29.0.

## 3. Results

### 3.1. Patients and Treatment Characteristics

A total of 245 patients were included in the study from 1 July 2017 to 30 June 2022. Baseline characteristics are summarized in Table 1. The median age was 59 years, with 61% of patients under the age of 65, and 58% were male. The majority of patients (82%) had a cutaneous melanoma primary site, and 44% of the patients presented BRAF V600 mutations. According to the 8th edition of the AJCC-TNM classification, 4% of patients were at stage IIB-C, 80% at stage IIIA-D and 16% at stage IV.

Treatment details are summarized in Table 2. The majority of patients (87%) received immune checkpoint inhibitor (ICPI) monotherapy, with nivolumab being the most common (74%). The median duration of treatment was 11.7 months [0.7–13.1 months]. In 78% of patients, adjuvant therapy began within 12 weeks of their last surgical resection. Reasons for treatment discontinuation were treatment completion (53%), relapse (32%) and toxicity (12%).

### 3.2. Survival and Prognostic Factors

The median follow-up period was 38.4 months (range 2–139, 95% CI: 35.7–41), with a minimum follow-up of 9 months for censored patients. Kaplan–Meier survival curves for RFS, DMFS, TTNT and OS are shown in Figure 1.

At data cut-off, 123 patients (50%) had presented recurrence. The median RFS was 33.7 months (95% CI: 20.7–46.4). Actuarial RFS rates at 18 and 36 months were 60% (95% CI: 54–66%) and 48% (95% CI: 40–56%), respectively. The median RFS according to different subgroups is presented in Table 3. In the univariate analysis, the primary site of melanoma, disease stage and interval from last resection to the start of adjuvant treatment were identified as predictors of RFS. In the multivariate analysis, a significant association was observed between RFS and both primary site of melanoma and time interval from last resection to initiation of adjuvant treatment.

At the time of data cut-off, 111 patients (45%) had developed distant metastases. The median DMFS was 43.7 months (95% CI: 30–59.4). Actuarial DMFS rates at 18 and 36 months were 65% (95% CI: 59–71%) and 52% (95% CI: 44–60%), respectively. The median DMFS according to different subgroups is presented in Appendix A. Primary site of melanoma was the only factor associated with DMFS.

At data cut-off, 113 patients (46%) had initiated additional systemic treatment with a median TTNT of 46.6 months (95% CI: 28.8–64.4).

Additionally, at data cut-off, fifty-six patients (23%) had died due to melanoma while three patients (1%) patients had died of unrelated causes. The median OS was not reached. Actuarial OS rates at 18 and 36 months were 89% (95% CI: 85–93%) and 75% (95% CI: 69–81%), respectively. The primary site of melanoma, the presence of ulceration and the stage of the disease were identified as predictors of OS. In the multivariate analysis, a significant association was found between OS and primary site of melanoma (Appendix A).

### 3.3. Toxicities

A summary of the observed toxicities is provided in Table 4. Serious adverse events (Grade 3 or higher) occurred in 11% of patients with no treatment-related deaths. Persistent toxicities were documented in 12% of cases, most commonly hypothyroidism (5%).

The presence of serious immune-related adverse events (irAEs) (grade 3 or 4) was also associated with longer RFS (median not reached [NR; 95% CI NR-NR] vs. 31.2 months [95% CI 20.9–41.5] if absence of irAEs, HR 0.4 [95% CI: 0.19–0.87]) and DMFS (median NR [95% CI NR-NR] vs. 34.2 months [95% CI 19.4–40] if absence of irAES, HR 0.4 [95% 0.18–0.94]). Figure 2 presents survival curves of patients with and without serious irAEs. No differences on RFS or DMFS were observed between different immunotherapies (*p* = 0.06) or between the use of monotherapy vs. combination (*p* = 0.22).

### 3.4. Patterns of Relapse

Among the 123 patients who experienced recurrence, 29 (24%) had locoregional relapse, 70 (57%) had systemic relapse and 24 (19%) had both locoregional and systemic relapse. The most common distant metastatic sites were distant lymph nodes (74 patients, 60%) and soft tissue (46 patients, 37%), while liver and brain metastases were present in 14 (11%) and 18 (15%) patients, respectively. The number of metastatic sites was less than three in 104 patients (85%).

Early relapse was observed in 78 patients (32% of all patients, 63% of patients with recurrence). Patterns of relapse according to time of relapse are shown in Table 5.

## 4. Discussion

The current retrospective multicenter observational study of patients treated with adjuvant immunotherapy for resected melanoma reports lower 18- and 36-month RFS, DMFS and OS rates and shorter median RFS and DMFS than those reported in clinical trials [7,8,9,10,11,12]. Negative prognostic factors for RFS were time from last resection to the start of adjuvant therapy (>12 weeks) and primary site of melanoma (especially mucosal melanoma). Serious immune-related adverse event (irAE) rates were consistent with those observed in clinical trials, and their presence were also associated with better RFS and DMFS. Additionally, early relapses presented with a higher number of metastatic sites than late relapses, although late relapses were more likely to present with brain metastases.

Our inferior survival results (compared to those in clinical trials) are in line with the recently reported 18-month RFS data in patients with stage III melanoma treated with adjuvant pembrolizumab in real-world setting [18]. Additionally, our study also provides information about DMFS, OS and 36-month RFS rates, which were also lower than those reported from clinical trials. Those findings support the relevance of real-world data to study the outcomes in an everyday practice population together with clinical trials.

One possible explanation for those lower outcomes may be the higher incidence of negative prognostic factors in real-world data studies compared with clinical trials. Interestingly, similar 12-month RFS rates have been reported when patients with similar characteristics to clinical trials are included [22]. In addition, similar OS to that reported in the Checkmate-238 trial has been reported in a real-world population, although RFS and DMFS data are lacking and the median follow-up for the real-world cohort of patients was 25.5 months, and data on OS data may not be mature enough [20].

The patients included in the present study have a higher median age than those of clinical trials and also include resected stage IV and mucosal melanoma in similar proportions to Checkmate-238, while these populations were excluded in the Keynote-054 trial [7,10]. In addition, this study included 22% of patients who started adjuvant therapy more than 12 weeks after their last resection, who were also excluded from clinical trials [7,10]. We did not confirm that stage was an independent prognostic factor, and the median RFS was similar in stage IV to that of the overall population, as observed in Checkmate-238 [10], but recent studies suggest the benefit of more aggressive treatment by adding ipilimumab to nivolumab in resected stage IV melanoma [16]. Our findings confirm the negative prognostic role of mucosal melanoma and suggest another subgroup of patients who may be candidates for treatment intensification with immunotherapy combinations or newer approaches, which should be explored in future clinical trials. Our analysis also evaluates the negative impact of delaying the start of therapy beyond 12 weeks after last resection, pointing out that efforts should be made in patient workflows to avoid delays in surgical recovery, oncologist referrals and start of therapy. In contrast, the presence of serious irAEs, although slightly lower than those reported in the clinical trials (14% in both Keynote-054 and Checkmate-238 [7,10]), was also associated with better RFS and DMFS, consistent with previous reports [23]. This evidence on prognostic factors may be relevant to adapt follow-up and define stratification factors in future studies.

Systemic relapse rates were similar to those previously reported, although early relapses were lower because more late relapses were detected with longer follow-up [19]. We also observed a higher incidence of brain metastases in patients with late recurrence, compared to early recurrences, emphasizing the importance of brain imaging in the follow-up of these patients.

This study has certain limitations inherent to its observational and retrospective nature, including selection bias, missing data and potential underreporting of relevant variables. In addition, follow-up studies were conducted according to clinical practice of each center and there may be differences. Patient follow-up, although longer than in previous real-world studies in this setting, might not be enough to properly assess OS outcomes and late recurrence patterns; also, the sample size may be small to adequately assess some prognostic factors. Last, the population may be heterogenous as it includes patients with stage IIB-C to IV, cutaneous and non-cutaneous melanoma, and treated with both anti-PD-1-based monotherapy and combinations. Despite these limitations, our study included a more representative population of clinical practice than clinical trials in terms of patient characteristics and treatment administration, and therefore complements other data from adjuvant clinical trials, but without the usual restrictions of a clinical trial. In addition, our results are consistent with previous real-world studies that provide further evidence on different patient subgroups and/or with longer follow-up.

## 5. Conclusions

The results of our real-world study showed that patients with resected melanoma treated with adjuvant anti-PD-1-based immunotherapy exhibited inferior outcomes in terms of recurrence-free survival (RFS), distant metastasis-free survival (DMFS) and overall survival (OS) compared to those observed in clinical trials, as reported in previous real-world data reports. The initiation of adjuvant therapy more than 12 weeks after the last resection and the presence of mucosal melanoma were associated with an increased risk of recurrence. Early relapses were more frequent than late relapses, although a higher incidence of brain metastases was observed in late relapses. These results support the need to investigate real-world data in parallel to clinical trials, and suggest that in patients with poor prognostic factors, new combination strategies should be investigated, in addition to closer follow-up.

## Figures and Tables

**Figure 1 cancers-17-00143-f001:**
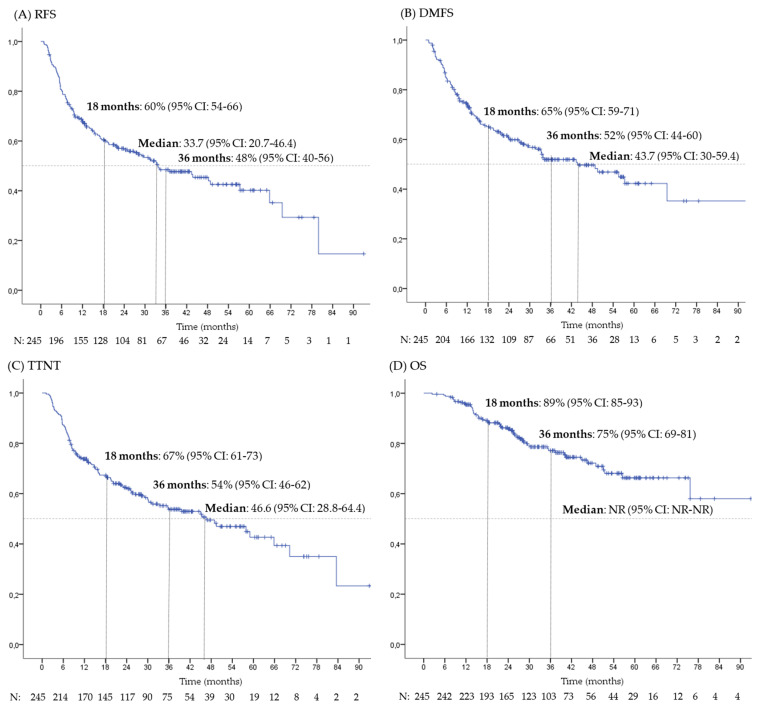
Survival outcomes. Kaplan–Meier curves for (**A**) recurrence-free survival (RFS); (**B**) distant metastasis-free survival (DMFS); (**C**) time to next treatment (TTNT); (**D**) overall survival (OS). Median times are reported in months. CI: confidence interval; NR: not reached.

**Figure 2 cancers-17-00143-f002:**
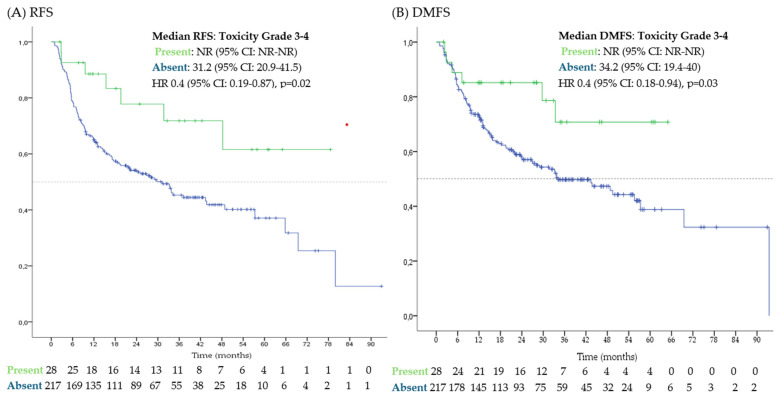
Survival outcomes according to the presence or absence of severe adverse events. Kaplan–Meier curves for (**A**) recurrence-free survival (RFS); (**B**) distant metastasis-free survival (DMFS). CI: confidence interval; HR: hazard ratio; NR: not reached.

**Table 1 cancers-17-00143-t001:** Baseline patients and disease characteristics.

	Patients (N: 245)
	N (%)/Median [Range]
Sex	
- Male	142 (58%)
- Female	103 (42%)
Age (years old)	59 [19–84]
- <65	150 (61%)
- 139	66 (27%)
- ≥75	29 (12%)
Primary site	
- Cutaneous melanoma	202 (82%)
- Acral melanoma	21 (9%)
- Mucosal melanoma	7 (3%)
- Unknown primary	15 (6%)
BRAF status	
- BRAF wild type	116 (47%)
- BRAF V600 mutation	108 (44%)
- Unknown	21 (9%)
Breslow index (mm)	3.7 [0–296]
- <0.8	20 (8%)
- 0.8–2	32 (13%)
- 5	44 (18%)
- 6	31 (13%)
- >4	103 (42%)
- Unknown	15 (6%)
Ulceration	
- Absent	87 (35%)
- Present	136 (56%)
- Unknown	22 (9%)
LDH (IU/dL)	186 [105–545]
- <ULN	111 (45%)
- ULN—2xULN	99 (40%)
- 2xULN–5xULN	29 (12%)
- >5xULN	6 (3%)
Mitosis (1/mm^2^)	
- <1	31 (13%)
- 11	118 (48%)
- ≥10	66 (27%)
- Unknown	30 (12%)
Stage (8th AJCC edition)	
- IIB	4 (2%)
- IIC	5 (2%)
- IIIA	15 (6%)
- IIIB	47 (19%)
- IIIC	120 (49%)
- IIID	14 (6%)
- IV	40 (16%)

AJCC: American Joint Committee on Cancer; IU: international units; LDH: lactate dehydrogenase; mm: millimeters; N: number of patients; ULN: upper limit of normality.

**Table 2 cancers-17-00143-t002:** Treatment characteristics.

	Patients
	N (%)/Median [Range]
ICPI Treatment	
- Anti-PD-1 monotherapy	214 (87%)
- Nivolumab	182 (74%)
- Pembrolizumab	23 (9%)
- Other	9(4%)
- Anti-PD-1-based combination	31 (13%)
- Ipilimumab–Nivolumab	15 (6%)
- Other combination	16 (7%)
Time on treatment (months)	11.7 [0.7–13.1]
Interval from last surgery initiation of adjuvant treatment	
<12 weeks	192 (78%)
>12 weeks	53 (22%)
End of treatment	
- Completion	129 (53%)
- Toxicity	30 (12%)
- Relapse	79 (32%)
- Ongoing	2 (1%)
- Unknown	5 (2%)

ICPI: immune checkpoint inhibitor; PD-1: programmed death-1; N: number of patients.

**Table 3 cancers-17-00143-t003:** Recurrence-free survival (RFS): median RFS and univariate and multivariate analysis according to patient and disease characteristics.

	RFS	Univariate HR	*p*-Value *	Multivariate HR
	Median (95% CI)	HR (95% CI)		HR (95% CI)
Sex				
- Male	37.1 (22.9–51.4)	Reference		
- Female	26.9–14.2–39.5)	1.13 (0.79–1.62)	0.49	
Age (years old)			0.39	
- <65	33.4 (25.7–41)	Reference		
- 65–74	79.9 (79.9-NR)	0.96 (0.66–1.39)	0.76	
- ≥75	15.4 (0–37.4)	1.47 (0.9–2.4)	0.12	
Primary site of melanoma			0.01	2.64 (1.15–6.01) *p* = 0.02
- Cutaneous melanoma	43.4 (26.1–60.7)	Reference		
- Acral melanoma	14.5 (6.3–22.6)	1.72 (0.96–3.08)	0.07	
- Mucosal melanoma	9.2 (1.8–16.6)	3.15 (1.37–7.25)	<0.01	
- Unknown origin	28 (6.2–49.7)	1.39 (0.7–2.76)	0.35	
BRAF status				
- BRAF wild type	33.2 (23.18–43.2)	Reference		
- BRAF V600 mutant	19.1 (3.1–35)	1.3 (0.9–1.84)	0.16	
Breslow index (mm)			0.81	
- <0.8	NR (NR-NR)	Reference		
- 0.8–2	43.4 (43.4-NR)	1.01 (0.91–2.15)	0.81	
- 5	33.8 (16.6–51)	1.25 (0.66–2.39)	0.5	
- 6	24 (0-NR)	1.32 (0.65–2.68)	0.44	
- >4	37.1 (15.4–58.9)	1.08 (0.6–1.93)	0.8	
Ulceration				
- Absent	57.3 (32.7–81.9)	Reference		
- Present	31.2 (19.9–42.5)	1.29 (0.87–1.92)	0.21	
LDH			0.47	
- <ULN	43.7 (26.6–60.8)	Reference		
- ULN—2xULN	19.5 (11.5–27.5)	1.37 (0.94–2)	0.11	
- 2xULN–5xULN	79.9 (49.2–110.7)	0.53 (0.27–1.06)	0.07	
- >5xULN	29.7 (16.1–43.3)	1.15 (0.79–1.66)	0.46	
Mitosis (/mm^2^)			0.96	
- <1	48.8 (48.7-NR)	Reference		
- 11	37.1 (14.8–59.5)	0.93 (0.52–1.64)	0.79	
- ≥10	33.8(19.1–48.4)	0.95 (0.51–1.77)	0.87	
Stage (8th AJCC edition)		1.5 (1.01–2.37)	**0.04**	1.5 (0.98–2.33) *p* = 0.06
- IIB-IIC	NR (NR-NR)	Reference		
- IIIA-IIIB	48.3 (29.8–66.7)			
- IIIA	43.4 (6.2–80.6)	2.66 (0.32–22.14)	0.37	
- IIIB	48.3 (24.7–71.8)	4.89 (0.65–36.51)	0.12	
- IIIC-IIIID	23.9 (11.7–36.2)			
- IIIC	24 (11.2–36.7)	7.06 (0.98–50.86)	0.05	
- IIID	19.1 (5.7–42.7)	7.01 (0.88–55.57)	0.06	
- IV	31.7 (20.6–42.7)	6.74 (0.91–49.77)	0.06	
Interval from last resection to initiation of adjuvant treatment				
<12 weeks	43.7 (31.5–55.9)	Reference		1.68 (1.13–2.5) *p* = 0.01
>12 weeks	14.7 (8.1–21.3)	1.64 (1.11–2.44)	0.01	

* *p*-values <0.05 are marked in bold. AJCC: American Joint Committee on Cancer; CI: confidence interval; HR: hazard ratio; IU: international units; LDH: lactate dehydrogenase; mm: millimeters; N: number of patients; NR: not reached; mRFS: median recurrence free survival; ULN: upper limit of normality.

**Table 4 cancers-17-00143-t004:** Immune-related adverse events (irAEs) from adjuvant anti-PD-1 therapy in patients with resected melanoma.

	Any Grade	Grade 3–4
	N (%)	N (%)
- No toxicity	135 (55%)	218 (89%)
- Any toxicity	110 (45%)	27 (11%)
Toxicity *		
- Cutaneous toxicity	76 (31%)	1 (<1%)
- Hypothyroidism	12 (5%)	0
- Colitis	10 (4%)	4 (2%)
- Arthritis	9 (4%)	1 (<1%)
- Hepatic toxicity	8 (3%)	7 (3%)
- Pneumonitis	8 (3%)	2 (1%)
- Nephritis	4 (2%)	3 (1%)
- Hypophysitis	1 (<1%)	1 (<1%)
- Pancreatic toxicity	1 (<1%)	1 (<1%)
- Adrenal insufficiency	1 (<1%)	1 (<1%)
- Gastritis	1 (<1%)	1 (<1%)
- Myositis	1 (<1%)	1 (<1%)
- Myocarditis	1 (<1%)	1 (<1%)
- Myelitis	1 (<1%)	1 (<1%)

* Patients may have more than one toxicity. N: number of patients.

**Table 5 cancers-17-00143-t005:** Melanoma recurrence patterns according to time and site of relapse after adjuvant therapy.

	All Patients with Relapse	Primary Resistance/Early Relapse	Late Relapse
	N (%)	N (%)	N (%)
	N = 123	N = 78	N = 45
Location of relapse			
- Locoregional relapse	29 (24%)	18 (23%)	11 (24%)
- Systemic relapse only	70 (57%)	40 (51%)	30 (66%)
- Locoregional and systemic	24 (19%)	20 (26%)	4 (9%)
Systemic relapse: affected organs *			
- Distant lymph nodes	74 (60%)	49 (63%)	25 (55%)
- Soft tissues	46 (37%)	34 (43%)	12 (26%)
- Bone	21 (17%)	12 (15%)	9 (20%)
- Lung and pleura	35 (28%)	20 (26%)	15 (33%)
- Liver	14 (11%)	12 (15%)	2 (4%)
- Brain	18 (15%)	7 (9%)	11 (24%)
- Others **	10 (8%)	8 (10%)	2 (4%)
Stage at recurrence			
- M0	29 (24%)	15 (19%)	11 (24%)
- M1a	50 (41%)	39 (50%)	14 (31%)
- M1b	12 (10%)	4 (5%)	8 (18%)
- M1c	16 (13%)	13 (17%)	3 (7%)
- M1d	16 (13%)	7 (9%)	9 (20%)
Number of metastatic sites			
- Less than 3	104 (85%)	64 (82%)	40 (89%)
- 3 or more	19 (15%)	14 (18%)	5 (11%)

* Patients may present more than one affected organ. ** Including peritoneal, adrenal and splenic metastases. N: number of patients.

## Data Availability

The raw data supporting the conclusions of this article will be made available by the authors on request.

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
