# Peer review of "Adjuvant Immunotherapy After Resected Melanoma: Survival Outcomes, Prognostic Factors and Patterns of Relapse"

_cancers, 2025, doi:10.3390/cancers17010143_

Round 1

Reviewer 1 Report

Comments and Suggestions for Authors

The manuscript discusses a retrospective multicenter observational study evaluating the outcomes of anti-PD-1-based adjuvant therapy in patients with resected melanoma. The authors investigate recurrence-free survival rates, prognostic factors influencing recurrence, and patterns of relapse. It compares these findings with clinical trial outcomes and highlights discrepancies in survival rates observed in routine clinical practice. I recommend accepting the manuscript with minor revisions to address the following points:

-The authors should consider expanding the introduction to provide a more comprehensive background on the global burden of melanoma and the significance of adjuvant immunotherapy in improving patient outcomes.

-Several sentences are lengthy and complex, making it difficult to follow the main point.

-More descriptive titles should be provided for Table 4 and Table 5.

-In the discussion, results are sometimes re-explained at length before interpretation. Focusing on interpretation within the discussion section would strengthen the section and improve clarity.

Author Response

Response to Reviewer Comments

1. Summary

Thank you very much for taking the time to review this manuscript. Please find the detailed responses below and the corresponding revisions/corrections in track changes in the re-submitted files.

2. Questions for General Evaluation

Reviewer’s Evaluation

Response and Revisions

Does the introduction provide sufficient background and include all relevant references?

Must be improved

This has been addressed in Comment 1.

Are all the cited references relevant to the research?

Yes

Is the research design appropriate?

Yes

Are the methods adequately described?

Yes

Are the results clearly presented?

Yes

Are the conclusions supported by the results?

Yes

3. Point-by-point response to Comments and Suggestions for Authors

The manuscript discusses a retrospective multicenter observational study evaluating the outcomes of anti-PD-1-based adjuvant therapy in patients with resected melanoma. The authors investigate recurrence-free survival rates, prognostic factors influencing recurrence, and patterns of relapse. It compares these findings with clinical trial outcomes and highlights discrepancies in survival rates observed in routine clinical practice. I recommend accepting the manuscript with minor revisions to address the following points

Comment 1: -The authors should consider expanding the introduction to provide a more comprehensive background on the global burden of melanoma and the significance of adjuvant immunotherapy in improving patient outcomes.

Response 1: We agree with this comment. As a result, the introduction has been expanded to provide more insights in the background.

-Introduction section, paragraph 1, lines 50-52: A new sentence has been added to explain melanoma global burden.

-Introduction section, paragraph 2, lines 57-69: this paragraph has been rewritten to add more information about clinical trials in adjuvant setting and the impact of adjuvant treatment on patient outcomes.

Comment 2: Several sentences are lengthy and complex, making it difficult to follow the main point.

Response 2: We thank the reviewer for this comment, which will help the article to be better understood. As a result, changes in response to comment 1 and comment 4 have been written with shorter sentences to make it easier to follow. In addition to the previously mentioned changes, we have also shortened some sentences at the following points:

-Introduction section, paragraph 4, lines 76 and 78

-Discussion section, paragraph 3, lines 253-254

Comment 3: -More descriptive titles should be provided for Table 4 and Table 5.

Response 3: We thank the reviewer for this suggestion. Therefore, we have changed the titles of Table 4 (Results section, line 201-202) and Table 5 (Results section, line 224-225).

Comment 4: -In the discussion, results are sometimes re-explained at length before interpretation. Focusing on interpretation within the discussion section would strengthen the section and improve clarity.

Response 4: We thank the reviewer for this comment, and we have done the following changes:

-Redundant results have been removed from the Discussion section at:

*Paragraph 1, lines 231 and 235-238;

*Paragraph 2, lines 241-246;

*Paragraph 4, lines 259-262, 276; and

*Paragraph 5, lines 281-282.

-Several comments and linkers have been added to the Discussion section to focus interpretation of findings at:

*Paragraph 1, lines 231-232 and line 238;

*Paragraph 2, lines 245-246 and 249-251;

*Paragraph 3, lines 253-255;

*Paragraph 4, lines 262, 263, 266, 267, 271, 274-275, 278-279; and

*Paragraph 5, lines 281-284.

4. Response to Comments on the Quality of English Language

Point 1:

Response 1: no comments on the Quality of English Language have been raised to be addressed.

5. Additional clarifications

In response to comment 1, references 1 and 2 have been added in the introduction and the order of references 3-15 has been changed. The number of lines in this response refers to the word document with track changes

Reviewer 2 Report

Comments and Suggestions for Authors

After reading the manuscript my major concerns are as follows:

  1. The authors should describe in more detail the results illustrated on Figure 2A and 2B. There was no information about a univariate hazard-ratio (HR) illustrated on Figures with its significant impact on Toxicity Grade 3-4. Although the HR values were statistically significant (p=0.02 and p=0.03), they were not described in the text, in contrast to multivariate HR, as presented in the Table 3 and discussed on page 6.
  2. The main question arises on how to perform statistical analysis with data reporting NR (not reached) values! Please, see the above-mentioned Figures 2A and 2B. How to compare the “present” data with their “absent” counterparts, if the “present” values were illustrated as NR with 95% Confidence Intervals (NR; NR)? This needs to be explained for the readers, who are not fluent in statistical analysis.

Author Response

Response to Reviewer Comments

1. Summary

2. Questions for General Evaluation

Reviewer’s Evaluation

Response and Revisions

Does the introduction provide sufficient background and include all relevant references?

Yes

Are all the cited references relevant to the research?

Yes

Is the research design appropriate?

Yes

Are the methods adequately described?

Must be improved

This has been addressed in comment 2.

Are the results clearly presented?

Can be improved

Are the conclusions supported by the results?

Can be improved

3. Point-by-point response to Comments and Suggestions for Authors

After reading the manuscript my major concerns are as follows:

Comment 1: The authors should describe in more detail the results illustrated on Figure 2A and 2B. There was no information about a univariate hazard-ratio (HR) illustrated on Figures with its significant impact on Toxicity Grade 3-4. Although the HR values were statistically significant (p=0.02 and p=0.03), they were not described in the text, in contrast to multivariate HR, as presented in the Table 3 and discussed on page 6.

Response 1: We thank the reviewer for pointing this out. In response to this comment, we have added the specific HRs in the result section, lines 207 and 208.

Comment 2: The main question arises on how to perform statistical analysis with data reporting NR (not reached) values! Please, see the above-mentioned Figures 2A and 2B. How to compare the “present” data with their “absent” counterparts, if the “present” values were illustrated as NR with 95% Confidence Intervals (NR; NR)? This needs to be explained for the readers, who are not fluent in statistical analysis.

Response 2: We agree with the reviewer that the description of this analysis can be improved. Although the median survival time is not reached in the “present” subgroup, the Kaplan-Meier does not just estimate median times, but considers the presence or absence of events at each interval time. This is true even if the number of events does not reach 50% and a median time cannot be reported. Since these survival analyses are considered informative if more than 10% of patients have the event, we have added an explanation in the Methods section, point 2.3, paragraph 3, lines 131-133.

4. Response to Comments on the Quality of English Language

Point 1:

Response 1: no comments on the Quality of English Language have been raised to be addressed.

5. Additional clarifications

The number of lines in this response refers to the word document with track changes

Round 2

Reviewer 2 Report

Comments and Suggestions for Authors

Thank you for explanation to the readers.

No further comments to the paper.

Author Response

We thank the reviewer for this response and the time and effort taken in reviewing the manuscript.